

# Assessing the temporally dynamic parameters in hydrological models: dynamic operations and evolutionary processes

Tian Lan[1,4], Kairong Lin[1,2,3], Chong-Yu Xu[4], Zhiyong Liu[2,3], Huayang Cai [3]

[1]Center of Water Resources and Environment Research, Sun Yat-sen University, Guangzhou, 510275, China.
[2]Guangdong Key Laboratory of Oceanic Civil Engineering, Sun Yat-sen University, Guangzhou, 510275, China.
[3]Southern Marine Science and Engineering Guangdong Laboratory (Zhuhai), 519000, China.
[4]Department of Geosciences, University of Oslo, P.O. Box 1047 Blindern, 0316 Oslo, Norway

*Correspondence to*: Kairong Lin (linkr@mail.sysu.eu.cn)

**Abstract.** The temporal dynamics of parameters can compensate for structural defects of hydrological models and improve
the accuracy and robustness of the streamflow forecast. Given the parameters usually estimated by global optimization
algorithms, a critical issue, however, which received little attention in the literature, is that the possible failure in finding the
global optimum might lead to unreasonable parameter values. This may cause the poor response of the dynamic parameters to
time-varying catchment characteristics (such as seasonal variations of land cover). In this regard, we propose a framework for
identifying the difficulty of finding the global optimum for dynamic hydrological model parameters by investigating their
evolutionary processes. Specifically, the probability distributions of the violin plots and the divergence measure of the
polylines in the parallel coordinates are applied and developed to configure the evolutionary processes in the individual
parameter spaces and multi-parameter space, respectively. Also, a complete solution for the dynamic operation of parameters
is proposed. Furthermore, clustering operations, calibration scheme and correlation between parameters are further discussed.
The results showed that the performance of the hydrological model with dynamic parameters achieves a significant
improvement. However, the response of individual parameters (even high-sensitive parameter) to dynamic catchment
characteristics is generally poor. The main reasons can be primarily attributed to the complexly linear and nonlinear correlation
between parameters and poor ability in finding the global optimum. In this regard, the dynamic parameter set instead of
individual dynamic parameters is suggested to extract dynamic catchment characteristics. Importantly, we found that the
properties of hydrological-model parameters, including identifiability, sensitivity, correlation and the ability to find global
optimum, interact with the response of parameters to the dynamic catchment characteristics. The ability to find global optimum
has a significant influence on the hydrological model performance with dynamic parameters. Hence, the ability to find global
optimum is suggested as one of the essential properties of the hydrological model parameters. The study provides a valuable
benchmark for temporally dynamic parameters in hydrological models.

## 1 Introduction

The dynamics of hydrological model parameters are usually used to overcome the structural inadequacy of models. For
example, dynamic components in hydrological models are oversimplified due to a poor understanding of their physical



mechanisms (Xiong et al., 2019; Dakhlaoui et al., 2017; Pathiraja et al., 2016;). Previous studies have demonstrated that the dynamics of parameters can effectively characterize the dynamic behavior of catchments and provided higher accuracy and robustness for streamflow forecast ( Dakhlaoui et al., 2017; Xiong et al., 2019; Motavita et al., 2019).

The efficient and effective estimations for dynamic parameters in hydrological models need to use optimization algorithms
due to the measurement limits and scale issues (Beven and Kirkby, 1979; Beven et al., 1984; Beven and Freer, 2001). Global optimization algorithms have been successfully applied in the automatic calibration of watershed models (Zhang et al., 2009; Gupta et al., 1998; Vrugt et al., 2005; Cooper et al., 1997). In particular, evolutionary algorithms (EAs) are the most well-established class of global optimization algorithms for solving water resources problems (Maier et al., 2014). However, Zhang et al. (2009) stated that the possible failure in finding the global optimum might lead to abnormal or unreasonable optimal
parameters. The unreasonable dynamic parameter values may not correspond well to the dynamic catchment characteristics, affecting the model performance with dynamic parameters (Sorooshian et al., 1993; Vrugt et al., 2005; Zhang et al., 2009). Hence, an identification of the ability to find global optimum is decisive for exploring the response of dynamic parameters on the time-variant dynamic catchment characteristics.

Maier et al. (2014) stated that the more the local optima, the more difficult it is to find the best solutions, which is the most
common cause of difficulty for finding the global optimum. The difficulty can be explained by comparing the evolutionary processes (Kallel et al., 1998). The fitness landscape is mainly an illustration of specific settings and states in the evolutionary processes ( Dawkins, 1997; Kauffman, 1993; Mitchell, 1998; Wright, 1932). As a conceptional and visualization tool, fitness landscape origins from theoretical biology and tries to solve real-world problems (i.e., globally optimal parameters in hydrological models) in evolutionary optimization problems. For hydrological models, the horizontal axes represent the
parameter space with all candidate solutions; The vertical axis represents the objective function values, i.e., the evaluations of a fitness function. To illustrate the overall structure of the fitness landscapes, such as the "big bowl" shape, can easily guide the algorithm towards the global optimum, while a surface that is tough with many local optima may present difficulties (Weise, 2009). The detailed information for the fitness landscape is provided in the Supporting Information.

Notably, the mapping of the fitness landscape for evolutionary processes is a challenge in hydrological model parameter
optimization. The main problems include highly non-linear, multimodal, non-convex, irregular, non-continuous, noisy, non-smooth and non-differentiable functions (Vrugt et al., 2005; Sorooshian et al., 1993; Gupta et al., 1998; Zhang et al., 2009). Notably, the hydrological simulation is not analytically derivable, which also increases the difficulty of the fitness landscape presentation (Maier et al., 2014). Several measures have been previously developed and applied to characterize the structure of fitness functions, including the correlation length (Weinberger, 1990), objective function surface (Duan et al., 1992,1993,
1994), fitness distance correlation (Jones and Forrest, 1995), the signal-to-noise ratio in the population sizing equation (Harik et al., 1999), the spatial autocorrelation statistic (Gibbs et al., 2004) and a dispersion metric (Arsenault et al., 2014). However, the simple method for the measurement of water resources problems still needs to be further explored. As developed in the





field of data visualization techniques, it is given the possibility to apply these state-of-the-art techniques to overcome the limitations of traditional techniques and explain new phenomena for the application of hydrological models, as well as to discover new insights (Arora and Singh, 2013; Derrac et al., 2014; Piotrowski et al., 2017; Gomez, 2019). In these regards, we develop a new framework for characterizing the structures of fitness landscapes with possible properties using geometry
visualization techniques and visualizing the evolutionary processes.

The purpose of this study is to give useful guidance for hydrological models with dynamic parameters. We give insights into assessing the temporally dynamic parameters in hydrological models, considering dynamic operations of parameters and the difficulty of finding the global optimum for dynamic hydrological model parameters by investigating the evolutionary processes. The rest of the paper is structured as follows. Section 2 presents data description and analysis as the case study
experiments; Section 3 presents the methods for investigating evolutionary processes. It also shows the approaches for estimating the dynamic hydrological model parameters. Section 4 presents the case study results; Section 5 discusses the dynamic parameters, quantitative evaluation of evolutionary processes, results of evolutionary processes, as well as outlines directions for future research; Section 6 summarizes the principal conclusions of the study.

## 2 Data description and analysis

Three basins are applied as an illustration in this study, as shown in Figure 1. The Hanzhong basin with 9,329 $km^2$ is located in the junction of Middle Yangtze basin, Wei River basin (i.e., the largest tributary to the Yellow River) and Chengdu Plain; low hills and moderate slopes characterize the Mumahe basin with 1,224 $km^2$; The Xunhe basin with 6,448 $km^2$ is dominated by a complex mountainous landscape, which has high temporal and spatial variability of soil moisture. Although the three basins with different rainfall-runoff characteristics, they all are located in the monsoon region of the East Asia subtropical
zone. It is cold and dry in winter but warm and humid in summer (Lin et al., 2010). The seasonal variations of vegetation density and types are contemporaneous (Fang et al., 2002). Significant seasonal changes in the climate and land-surface conditions allow for exploring the intra-annual dynamics of the hydrological processes. Daily streamflow and climatic data from 1980 to 1990 were used. Nearly 73% of the data samples (1980-1987) without significant human- interference (i.e., no little of reservoir regulations) were used for calibration, and the remainder (1988-1990) was utilized to verify the model.

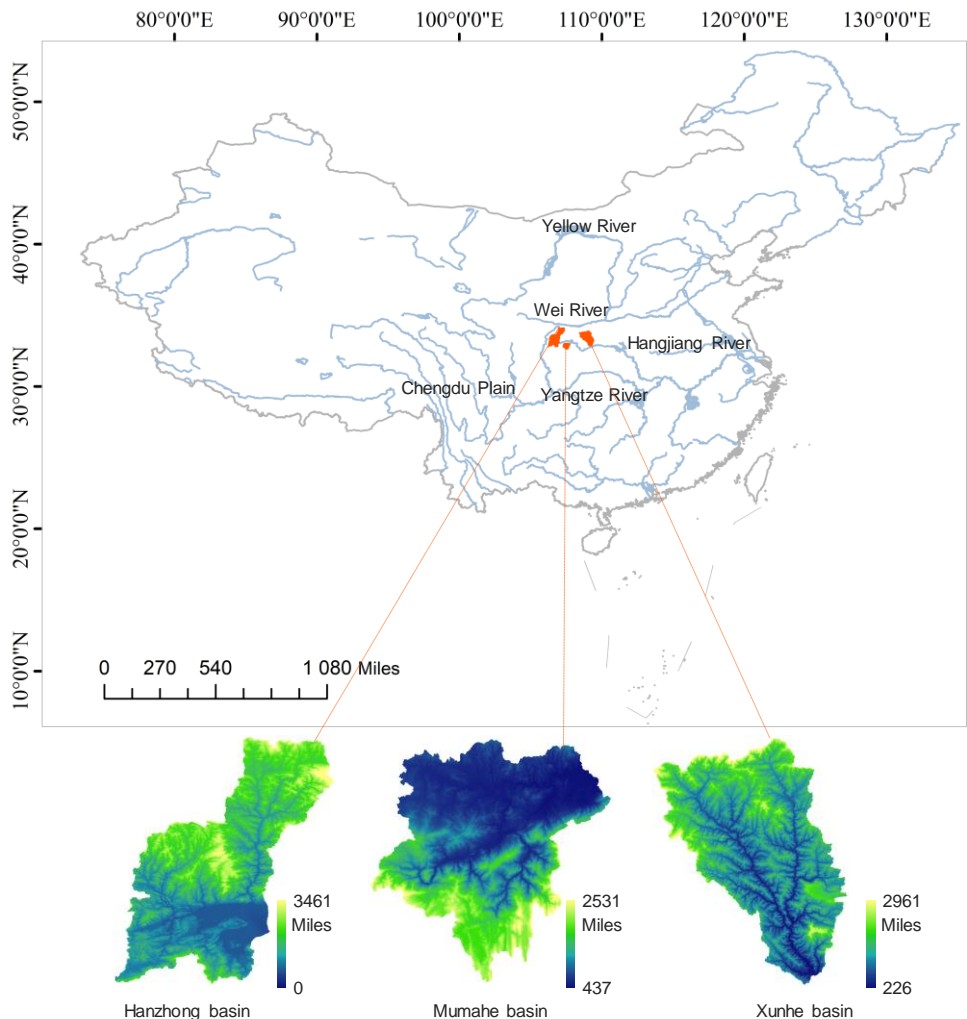

**Figure 1.** Locations of the study region.

# 3 Methodology

## 3.1 Dynamic operations of parameters

### 3.1.1 Extraction of dynamic catchment characteristics

A set of climatic-land surface indices was provided and pre-processed using the maximal information coefficient (MIC) and principal components analysis (PCA). MIC is used to identify the intricately linear and nonlinear correlation between indices and screen the indices (Zhang et al., 2014). PCA is applied to reduce the dimensionality of indices (Minaee et al., 2018). The climatic indices included total precipitation, maximum 1-day precipitation, maximum five-day precipitation, moderate precipitation days, heavy precipitation days, total pan evaporation, maximum 1-day pan evaporation and minimum 1-day pan





evaporation. The land-surface indices included antecedent streamflow and runoff coefficient. Two clustering operations were performed based on the pre-processed climatic and land-surface index systems, respectively. Namely, the calibration period is partitioned into different sub-periods based on the climatic and land-surface indices, respectively. More information is elaborated in Lan et al. (2018). The flowchart is illustrated in Figure 2 and its codes are opened (If you have interests in codes,

please do not hesitate to contact us).

### 3.1.2 Calibration for dynamic parameters

In the calibration period, only the data from the individual sub-periods are used for minimizing the objective function, while the model is run for the whole period. The simulated flow data from each sub-period are then combined and compared with the observed flow (Lan et al., 2020) (see Figure 2). Calibration has a warm-up period of one year and three months in the

validation period.

HYMOD model is one of the commonly used lumped rainfall-runoff models (Yadav et al., 2007; de Vos et al., 2010; Pathiraja et al., 2018). It is selected as illustration purposes in this study. The definitions of the five parameters are illustrated in Table 1. HYMOD mainly includes soil moisture accounting mode (involving parameters $H_{uz}$, $B$ and alpha) and flow routing mode (involving $K_q$ and $K_s$). The more detailed descriptions are presented in Supporting Information.

**Table 1.** Definitions of parameters used in the HYMOD model (Wagener et al., 2001).

| Label | Property | Range | Description |
|-------|----------|-------|-------------|
| $H_{uz}$ | Parameter | 0-1000 [mm] | Maximum height of soil moisture accounting tank |
| $B$ | Parameter | 0-1.99 | Scaled distribution function shape |
| alpha | Parameter | 0-0.99 | Quick/slow split |
| $K_q$ | Parameter | 0-0.99 | Quick-flow routing tanks' rate |
| $K_s$ | Parameter | 0-0.99 | Slow-flow routing tank's rate |

An evolutionary algorithm for dynamic parameters used in this study is the so-called Shuffled Complex Evolution from the University of Arizona (SCE-UA) (Duan et al., 1993). Arsenault et al. (2014) demonstrated that SCE-UA performs better for hydrological models with low complexity, compared with other global optimization algorithms. Besides, the multiple trials

are performed to ensure that the results are consistent, preventing the effects of initial values. The objective function is defined as the combination of the Nash-Sutcliffe efficiency index (NSE) and its logarithmic transformation (LNSE) (Nash and Sutcliffe, 1970; Nijzink et al., 2016). It is expressed as $1 - 0.5 \cdot (NSE + LNSE)$. The closer the objective function value is to zero, the better the model performance.

### 3.1.3 Multi-metric assessment of streamflow simulation

Simulation performance with dynamic parameters is assessed using seven performance metrics. The metrics include NSE, LNSE and a five-segment flow duration curve (5FDC) with the root mean square error (RMSE) (Pfannerstill et al., 2014). The NSE is sensitive to peak discharges and LNSE emphasizes low flows. RMSE with FDC is used to assess the model performance





in the five phrases of streamflow, including very high, high, middle, low and very low flow (Cheng et al., 2012; Pokhrel et al., 2012; Yokoo and Sivapalan, 2011). FDC is split into five segments, including below Q5, between Q5 and Q20, between Q20 and Q70, between Q70 and Q95, and higher than Q95, i.e., RMSE_Q5, RMSE_Q20, RMSE_Qmid, RMSE_Q70, RMSE_Q95.Besides, the differences in these metrics between the calibration period and the validation period are used to

5 assess the temporal transferability of parameters (Gharari et al., 2013; Klemeš, 1986) (see Figure 2).

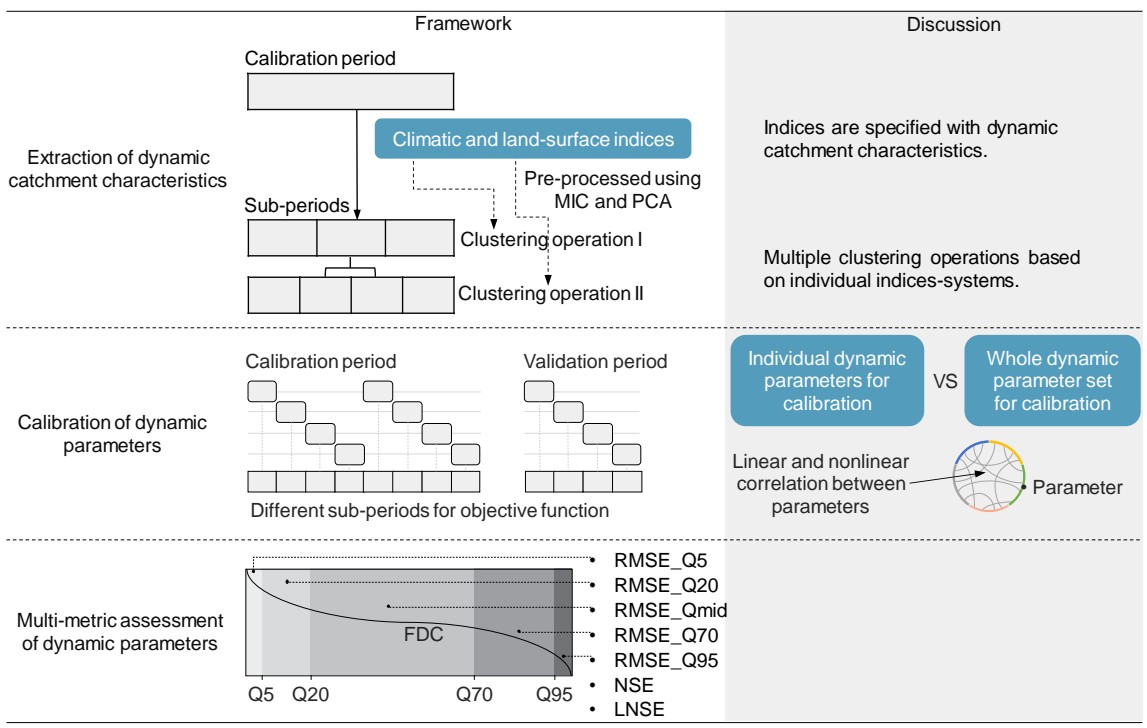

**Figure 2.** The developed framework for dynamic operations of parameters.

### 3.2 Investigation of evolutionary processes

#### 3.2.1 Theory

10 In each evolutionary process, four steps, including evaluation, fitness assignment, selection and reproduction, are performed. The parameter set with the best objective function value in each evolutionary process loop is recorded in the "evolutionary processes". The final optimum is obtained at the end of the run while satisfying the stopping criteria. The evolutionary process evolves toward minimizing the objective function values. Notably, each loop has numerous objective function evaluations ( Zheng et al., 2017; Azad, 2019; Gomez, 2019). If the corresponding points of all the objective function values are used, these

15 semi-random points cannot effectively visualize and investigate the evolutionary process.

Fitness landscapes are a potent metaphor for visualizing the evolutionary processes in evolutionary algorithms. The intuitive sketches of three-dimensional fitness landscapes with possible properties are illustrated in Figure 3a. The vertical axis denotes





the objective function values and the horizontal axes denote the parameter space. The possible properties with increasing difficulty to find global optimum are illustrated as follows:

(I) Best case or low variation: an evolutionary process is ideal for estimating the global optimum.

(II) Deceptiveness: a significant obstacle is a local optimum. The gradient of the deceptive objective function values may lead the optimizer away from the optima.

(III) Confusion: as the local optima increase, the possible paths for finding the globally optimal solution are complicated, which makes it harder to find the global optimum.

(IV) Ruggedness: if the objective function values are fluctuating, i.e., increasing or decreasing, it is difficult to determine the correct direction for the evolutionary process (Weise, 2009).

### 3.2.2 Individual parameter spaces

The violin plots are a tool to visualize the kernel density distribution of the data points (Hintze and Nelson, 1998; Piel et al., 2010). The anatomy of the violin plot and the associated information can be found in the Supporting Information (section 1.3). We use probability distributions of the violin plots to configure the elements of the evolution processes, representing the possible properties of the fitness landscapes.

As shown in Figure 3b, the vertical axis of the violin plot denotes parameter values; the horizontal axis denotes the probability values. With an adequate parameter space and sufficient density of coverage in individual parameters, the thinner distribution type of violin plot indicates that fewer local optimal solutions hamper evolutionary processes. For example, unimodal distribution is an ideal evolutionary process to estimate the best solution. Conversely, the multimodal or flat distribution signifies that the search is indecisive due to the prominent interference from local optima. Namely, the search may fail to find a global optimum (Dakhlaoui et al., 2017; Rahnamay Naeini et al., 2018; Vrugt and Beven, 2018). The four types of distributions of violin plots (including unimodal, bimodal, multimodal and flat distributions) with increasing the number of peaks match the properties' sketches of the fitness landscapes.





**Figure 3. a,** Intuitive sketches of three-dimensional fitness landscapes with possible properties. The vertical axis denotes the objective function values and the horizontal axes denote the parameter space. The arrows represent various paths that the population could follow while evolving on the fitness landscape. **b,** Evolutionary processes in individual parameter spaces using violin plots. The vertical axis of the violin plot denotes parameter values; the horizontal axis denotes the probability values. **c,** Evolutionary processes in multi-parameter space using parallel coordinates. These parallel axes represent individual parameters. The polylines describe the parameter set. The evolutionary process evolves toward minimizing the objective function values $f$(x). Hence, the color changes of parallel coordinates could represent the evolutional direction of the fitness landscapes. **d,** Multi-relational 3D parallel coordinate plots. **e,** Spotting the envelope of lines between adjacent axes to scatter plot.

### 3.2.3 Multi-parameter space

The evolutionary process of dynamic parameters in multi-parameter space is investigated (see Figure 3c). The parallel coordinates are a data visualization technique for multivariate data that is easy to interpret, which are applied to configure the evolutionary processes in the multi-parameter space. The polylines describe multivariate items that intersect with parallel axes. These parallel axes represent variables that can be used for the analysis of multiple properties of a multivariate data set (Heinrich and Weiskopf, 2015; Janetzko et al., 2016; Johansson and Forsell, 2016). More detailed information on the parallel coordinates is given in the Supporting Information. When used in hydrological models, the variables on the dimension axes denote individual parameters. The polylines of the parallel coordinates symbolize the parameter sets in all loops of one evolutionary process. The divergence measure of the polylines in the parallel coordinates is used to assess the ability to find global optimum in the multi-parameter space. The higher the divergence, the more difficult it is to determine the correct direction for the evolutionary process. Accordingly, the parameter set is challenging to converge to the global optimum. Moreover, the evolutionary process evolves toward minimizing the objective function values $f$(x). Hence, the color changes of parallel coordinates (see Figure 3c) could represent the evolutional direction of the fitness landscapes, which is illustrated in Figure 3a (III). The direction of the arrow represents the direction of evolution. Also, the violin plot can visualize the probability distribution of each variable (i.e., parameter) along the dimension axis (Janetzko et al., 2016) (see Figures 3b and c). Interestingly, in the axis configuration of parallel coordinates, the linear relationships between variables mapped on adjacent axes can be directly analyzed (Vrotsou et al., 2010). In this regard, the multi-relational 3D parallel coordinates (Yao and Wu, 2016) (see Figure 3d) might be applied to analyze the correlation between any two parameters and explore new phenomena for a run of hydrological models in the ongoing research. Interestingly, the envelope of lines between adjacent axes can be spotted to the scatter plot, which represents the relationships between parameters, as shown in Figure 3e. In this regard, the potential nonlinear and linear correlation between parameters is illustrated by the MIC metric and the Pearson correlation coefficient $r$.





# 4 Results

## 4.1 Dynamic parameters

### 4.1.1 Dynamic catchment characteristics

The calendar year is divided into four sub-annual periods based on hydrological/climatic similarities, as shown in Figure 4a.

In this way, the clustering results of the validation period are largely in agreement with the results of the calibration period. The sub-periods include the dry period, rainfall period I, rainfall period II (wettest period) and rainfall period III. Both the total amount and the variance values of all the precipitation series are minimum in the dry period and maximum in the rainfall period II. Two normal sub-annual periods (rainfall period I and rainfall period III) have similar climate conditions, but the rainfall period III has higher antecedent soil moisture content than in rainfall period I.

### 4.1.2 Dynamic parameter set

Dynamic parameter sets in study areas are shown in Figure 4b. The value of $K_s$ (slow-flow routing tanks' rate) is lowest in the dry period and highest in the wettest period in all basins. However, other parameters have no regular pattern on dynamic catchment characteristics. Most of the excess streamflow in the three rainfall periods is diverted to the slow-flow tank because the alpha values are close to zero. It means that the slow-flow tanks have a primary effect on the simulations. The parameter

$K_s$ does not reflect catchment characteristics in rainfall period I and rainfall period III, which have similar climate conditions but not land-surface conditions. In sum, there is generally poor response of dynamic parameter set to catchment dynamics.

### 4.1.3 Multi-metric evaluation

The model performance is presented in Figure 4c. The results show that the model with dynamic parameters has significant improvement in five phases (i.e., very high, high, middle, low and very low flows) of streamflow in the calibration period and

validation period. Moreover, the reduction of gaps of metric values between calibration period and validation period demonstrates better temporal transferability of the dynamic parameters. Also, the simulation performance in four sub-periods of the calibration period is shown in Figure 4d. The results indicate that the model performance is best in the rainfall period II (wettest period) and the poorest in the dry period.



**Figure 4. a,** Heat map of sub-period partition. **b,** Dynamic parameter sets. **c,** Model performance. **d,** Simulation performance in four sub-periods of the calibration period.





### 4.2 Evolutionary processes

#### 4.2.1 Individual parameter spaces

Taking the Hanzhong basin as an example, the results for investigating the evolutionary processes of dynamic hydrological model parameters in the individual parameter spaces are shown in Figure 5. The interpretation is as follows: Considering the

different sub-periods, in the dry period, except for $K_s$, the distributions of violin plots in other parameter spaces are oscillating in the entire feasible parameter space. However, the magnification of the parameter $K_s$ also shows multimodal distributions. The results imply that the final values in these parameter spaces may have been affected by the considerable local optima. Namely, the ability to find global optimum in all parameter spaces in the dry period is generally poor. In the rainfall periods I, II and III, the parameters $\alpha$ and $K_s$ with unimodal distribution have a higher ability to find the global optimum. Other

parameters with bimodal or multimodal distributions present a more inferior ability to find the global optimum. In sum, the ability to find optimum in three rainfall periods is higher than the ability in the dry period.

Considering different parameters, the parameter $K_s$ presents the thinner distribution of violin plots in all sub-periods, which manifests that its evolutionary processes are disturbed by fewer local optima. However, the ability to find the global optimum of other parameters is generally poor. The results in the Mumahe basin and Xunhe basin are shown in Figure S3 and Figure

S4 in the Supporting Information. The results are consistent with those of the Hanzhong basin.

#### 4.2.2 Multi-parameter space

The evolutionary processes of dynamic parameters in multi-parameter space are investigated and shown in Figures 6 and 7. Firstly, it is stated that the parameter sets from the first two loops are not investigated because their results have high uncertainties as to the warm-up of the global optimization algorithm. The interpretation is as follows: The divergence measure

of the polylines decreases sequentially in the dry period, rainfall period I, rainfall period III and rainfall period II. The result indicates that the ability to find global optimum is increasing in four sub-periods.





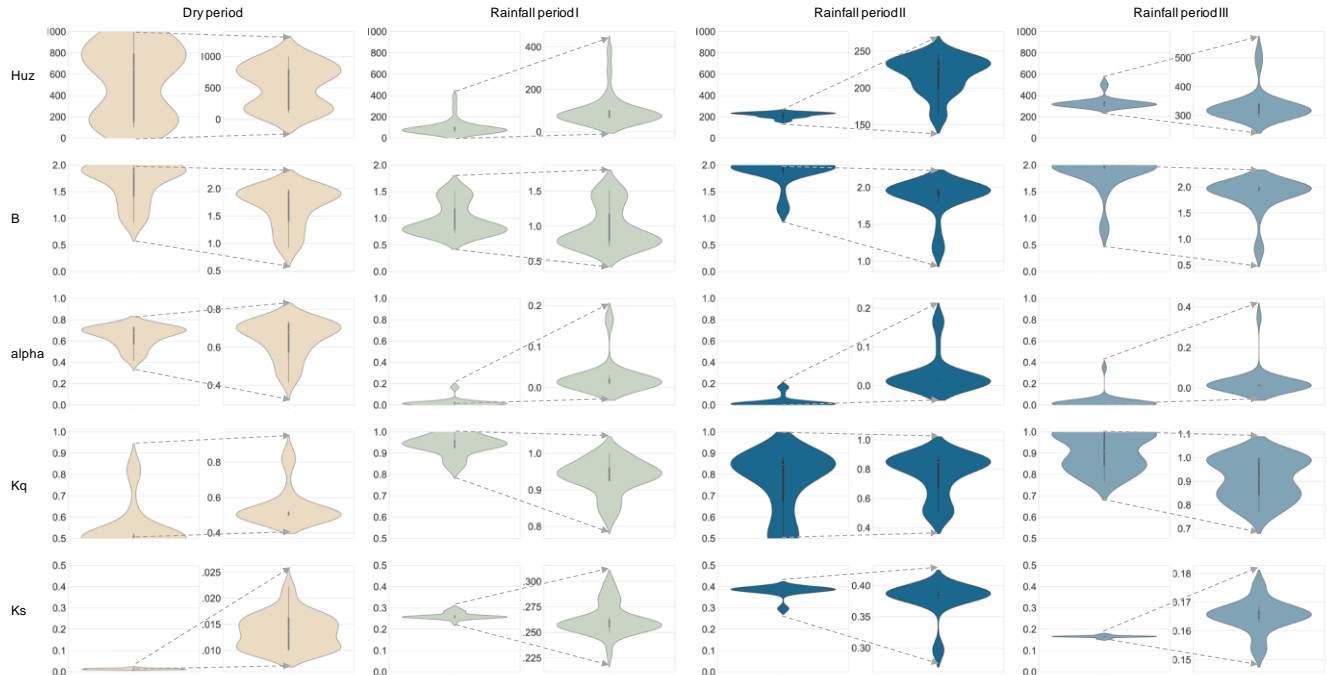

**Figure 5.** Evolutionary processes of dynamic parameters in the individual parameter spaces in the Hanzhong basin.

The direction of the evolutionary processes in the multi-parameter space is analyzed according to the color changes of parallel
coordinates (see Figures 6 and 7). In the dry period, the objective functions corresponding to the polylines are fluctuating, i.e.,
increasing and decreasing. It implies that it is difficult to find the right direction to determine the global optimum. In the rainfall
period II, the objective function corresponding to the polylines searches to the final values with the minimum number of
iterations, i.e., the fastest speed. These results not only verify the results mentioned above but also present the dynamic
evolutionary behavior with evolutionary direction. The results in the Mumahe basin and Xunhe basin are similar to the
Hanzhong basin and shown in Figures S5-S8 in the Supporting Information.





**Figure 6.** Evolutionary processes of dynamic parameters in multi-parameter space in the Hanzhong basin.



**Figure 7.** Evolutionary processes of dynamic parameters with magnified details on the axes in multi-parameter space in the Hanzhong basin.





## 5 Discussion

### 5.1 Dynamic parameters

For the estimation of dynamic parameters, the indices should be specified with dynamic catchment characteristics. The climate and land-surface indices were selected as examples in this study. Notably, the clustering results represent the relative
differences of the sub-annual periods in a basin rather than absolute differences. Moreover, it was demonstrated that the model performance was better when two sub-period clustering operations were performed based on climate indices and land-surface indices, respectively, instead of one clustering operation according to all indices. The reason was that according to all indices, the unsupervised clustering method might not well identify the main characteristics of sub-periods in various systems. For example, the rainfall period I and rainfall period III with similar climate conditions but not land-surface conditions might not
be distinguished.

The sub-period calibration based on extracted dynamic catchment information is one of the simplest approaches to achieve dynamic parameters. More techniques for dynamics of parameters have been developed (Xiong et al., 2019; Motavita et al., 2019; Manfreda et al., 2018; Lan et al., 2018; Fowler et al., 2018; Fowler et al., 2018). A specific issue was discussed whether the dynamic of a single parameter with high sensitivity or identification could significantly improve the simulation
performance of hydrological models. We explore this issue as follows. First, a simple but useful tool, i.e., a scatter plot, is used for identifying the sensitive parameter of hydrological models (Paruolo et al., 2013). As shown in Figure 8a, the parameter $K_s$ of HYMOD has the highest sensitivity in the three study areas. Then, the parameter $K_s$ in the different sub-periods and other fixed (time-invariant) parameters are optimized simultaneously during one run in this study. Take the Hanzhong basin as an example, the results (see Table A1 in the Appendix) indicate that the model performance is barely improved than a model run
with time-invariant parameters.

Furthermore, Bárdossy (2007) demonstrated that changes in one parameter might be compensated for by changes in other parameters due to their interdependence (Westra et al., 2014; Klotz et al., 2017; Wang et al., 2017, 2018). Therefore, although a specific parameter is dynamic, the other parameters may counteract those changes due to their complex correlations, resulting in no overall change in the hydrological processes. The complex correlation between parameters in three study areas was
calculated to verify the Bárdossy's (2007) view using the maximal information coefficient (MIC). MIC metric quantifies the linear and nonlinear correlations between parameters (Albanese et al., 2018; Kinney and Atwal, 2014; Zhang et al., 2014). Figure 8b shows that the MIC values are generally high in study areas, which demonstrates that the significant linear and nonlinear correlation existed between parameters.





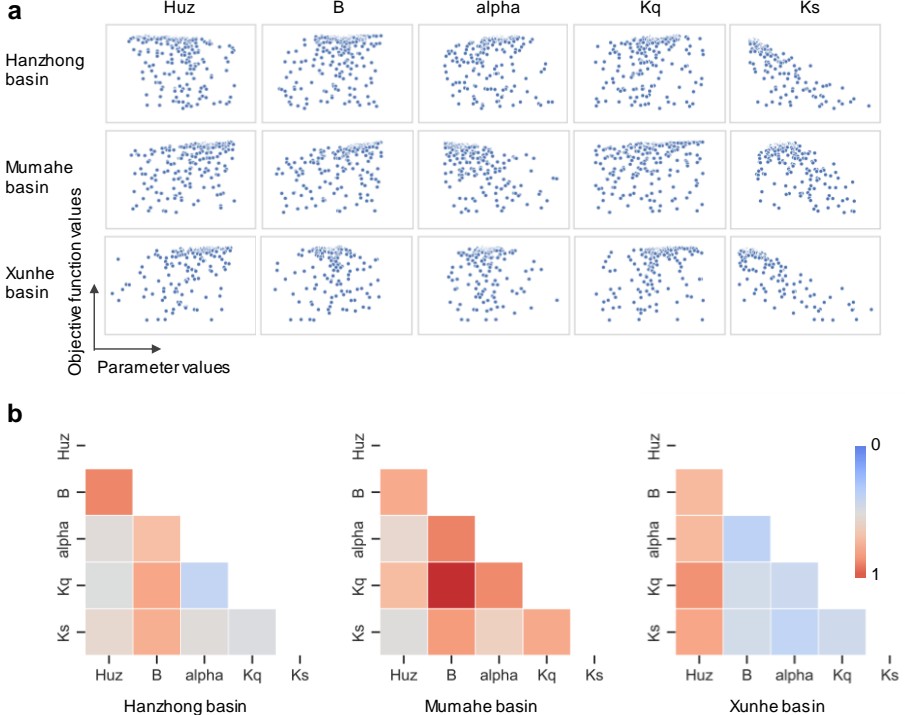

**Figure 7. a,** Sensitivity analysis results using scatterplots. The horizontal axis represents the sampling points, which are the parameter sets; The vertical axis represents their objective function values. **b,** The linear or nonlinear correlations between the parameters based on MIC coefficients. Red denotes the strongest correlation between parameters.

## 5.2 Quantitative evaluation of evolutionary processes

One important issue with this study is the use of quantifiable metrics to assess the evolutionary processes. The investigation and visualization techniques above are analyzed. (1) The violin plot uses a nonparametric density estimation based on a smooth kernel function with a fixed global radius. Therefore, the PDF (probability density function), CDF (cumulative distribution function) values of data can be used to quantify the violin plots in case of uniform, multimodal, skewed and clipped data (Yapo et al., 1996). The mathematical benchmark functions with PDF and CDF will be used for assessing the evolutionary processes of dynamic parameters in the next research. (2) A dispersion metric is suggested to evaluate the polylines in the parallel coordinate and the evolutionary processes in the multi-parameter space. The metric measures average Euclidian distances, which were normalized to ensure comparability (Arsenault et al., 2014). However, the application of the quantitative evaluation metrics needs a significant amount of experiment, validation, analysis and discussion, which cannot all be considered in this study. We will clarify and investigate this critical issue in the ongoing study.



## 5.3 Evolutionary processes

Considering different sub-periods in individual parameter spaces, the ability to find global optimum in three rainfall periods is higher than the ability in the dry period. The results are consistent with the evaluation results of simulation performance in different sub-periods. Namely, compared with the dry period, streamflow simulation exhibits better performance in the three
rainfall periods. The main reason is that the HYMOD model is well suited for watersheds dominated by saturation-excess overland flow processes (Sarrazin et al., 2016; Pathiraja et al., 2018; Wang et al., 2018). Intense rainfall events mainly contribute to saturation-excess overland flow in the rainfall periods (Wi et al., 2015; Sarrazin et al., 2016). Consequently, the ability to find global optimum has a significant influence on the hydrological model performance. Notably, besides the common adverse effects of the ability to find global optimum in hydrological models, such as dataset errors, nonlinear nature of models
and premature convergence, the applicability of models to variant catchment characteristics is also a critical effect for the evolutionary processes in hydrological models.

Considering different parameters in individual parameter spaces, except the parameter $K_s$ with the highest sensitivity, the ability to find the global optimum of other parameters is generally poor. The results are consistent with the evaluation results of dynamic parameters. Namely, only the parameter $K_s$ is lowest in the dry period and highest in the wettest period in all basins.
Other parameters present the poor response to dynamic catchment characteristics. The reasons are that the $\alpha$ values in the three rainfall periods are close to the minimum, which indicates that the slow-flow tank controls the hydrologic model responses. Hence, the parameter $K_s$ is the critical parameter for streamflow simulation. Moreover, the results also illustrate that the cascade routing component parameterized by $\alpha$ and $K_q$ (or $K_s$) in the model system has stronger dominance than the soil moisture component parameterized by $H_{UZ}$ and $B$. Consequently, the critical parameter $K_s$ has not only the highest sensitivity
and identifiability but also the best ability to find the global optimum. These properties interact with the response of parameters to the dynamic catchment characteristics. Notably, the ability to find global optimum is suggested as one of the essential properties of the hydrological model parameters, such as identifiability, equifinality, sensitivity, correlation, and uncertainty, while evaluating the model parameters.

Considering different sub-periods in multi-parameter space, the ability to find global optimum in multi-parameter space
increases in the dry period, rainfall period I, rainfall period III and rainfall period II. Also, it is difficult to find the right direction to determine the global optimum in the dry period but most accessible in the rainfall period II. The results in multi-parameter space not only verify the results in individual parameter spaces but also show the direction and speed of evolution.

Importantly, the response of all parameters to dynamic catchment characteristics is generally poor, although the parameter $K_s$ have a response to dynamic climate conditions but not land-surface conditions. However, according to multi-metric evaluation
results, the model performance with a dynamic parameter set has significant improvement in high, middle, and low phases of streamflow. The increase in medium flow mainly benefits from the extraction of dynamic land-surface information. The clustering of the rainfall period I and rainfall period III is based on diverse soil moisture content but similar climate conditions.





Besides, there was better temporal transferability of the dynamic parameters in the calibration and validation periods. Consequently, sub-period calibration can improve the hydrological model performance via extracting the dynamic catchment characteristics. Even though individual parameters cannot respond well to dynamic catchment characteristics due to the correlation between parameters, a dynamic parameter set can be as an external framework by extracting dynamic catchment

characteristics and improve the model performance.

## 5.4 Limitation

Still, there are several limitations, and we will address in future studies (1) More catchments with various characteristics will be investigated to explore the impact of spatial variability of watershed features on evolutionary processes and model performance. (2) Besides seasonal-scale variability, more time scales for dynamic parameters in catchment response, such as

annuals-scale variability and long-term changes, will be further studied. (3) The study uses a 5-parameter model, which is considered as a small parameter space. We would explore the higher-dimensional hydrological model using the methodology and procedure demonstrated in this study. (4) More discussions will be provided to investigate the evolutionary processes in dynamic hydrological model parameters, including the size of the search space and the relative size of the feasible space. (5) The performance of the proposed strategy has been demonstrated in the context of single-objective optimization problems. To

account for the multi-objective nature of the model calibration, the discussion on the potential extension of the proposed framework to the multi-objective optimization and the notion of Pareto front will be carried out. (6) The evolutionary speed will be explored in ongoing research.

## 6 Conclusions

This study gives new insight into assessing the dynamic operations of hydrological-model parameters and their evolutionary

processes. The main conclusions can be drawn:

The extraction of dynamic catchment characteristics is via clustering operations based on pre-processed the climatic and land-surface indices, respectively. It was demonstrated that the model performance was better when two sub-period clustering operations were performed, respectively, instead of one clustering operation according to all indices. Moreover, the effect of the dynamics of a single parameter with high sensitivity or identification on individual parameters is investigated. The results

showed that the performance of the model with a single dynamic parameter is barely improved than a model run with time-invariant parameters. The main reason was that although a specific parameter is dynamic, the other parameters may counteract those changes due to their complex correlations, resulting in no overall change in the hydrological processes.

The fundamental theories for evolutionary algorithms and fitness landscapes were introduced and analyzed. The probability distributions of the violin plots are applied and designed to configure the elements of the evolution processes, representing the

possible properties of the fitness landscapes in the individual parameter spaces. The divergence measure of the polylines in the



parallel coordinates was developed to configure the evolutionary processes in the multi-parameter space, representing the evolutional direction of the fitness landscapes. The quantitative metrics, PDF (probability density function), CDF (cumulative distribution function), and dispersion metric were preliminarily discussed for assessing the ability to find the global optimum of dynamic hydrological model parameters.

According to the results of assessing the evolutionary processes of dynamic parameters, we found that the ability to find global optimum has a significant influence on the hydrological model performance with dynamic parameters. The properties of parameters, including identifiability, sensitivity, correlation and the ability to find global optimum, interact with the response of parameters to the dynamic catchment characteristics. Hence, the ability to find global optimum is suggested as one of the essential properties of the hydrological model parameters. Moreover, the applicability of models to variant catchment

characteristics also has a critical effect on the evolutionary processes in hydrological models. The results in multi-parameter space not only verify the results in individual parameter spaces but also show the direction and speed of evolution.

The performance of the hydrological model with dynamic parameters has a significant improvement in low, middle and high flows, as well as better temporal transferability of the dynamic parameters. However, the response of all parameters to dynamic catchment characteristics is generally poor. Consequently, even though individual parameters cannot respond well to dynamic

catchment characteristics due to the correlation between parameters, a dynamic parameter set can be as an external framework by extracting dynamic catchment characteristics and improve the model performance.

We hope that the results of this study provide valuable information that can be applied to improve our understanding of hydrological processes and dynamic hydrological model parameters.

**Acknowledgments**

This study is financially supported by the Excellent Young Scientist Foundation of NSFC (51822908), the National Natural Science Foundation of China (No. 51779279), the National Key R&D Program of China (2017YFC0405900), Open research foundation of Dynamics and the associated process control key laboratory in the pearl river estuary of ministry of water resources (2017KJ12), Baiqianwan project's young talents plan of special support program in Guangdong Province (42150001), and the Research Council of Norway (FRINATEK Project 274310). Digital Elevation Model (DEM) of the study area is

derived from the Advanced Spaceborne Thermal Emission and Reflection Radiometer (ASTER) global digital elevation model (GDEM) with a cell size of 30 × 30 m which are obtained from https://asterweb.jpl.nasa.gov/. The climatic datasets consist of daily rainfall datasets and pan evaporation datasets provided by the China Climatic Data Sharing Service System which are obtained from https://data.cma.cn/en. Daily streamflow used to support this paper can be made available for interested readers by the corresponding author at linkr@mail.sysu.edu.cn.





## Appendix

**Table A1.** Performance for HYMOD model with time-invariant parameters (Scheme 1) and model with a dynamic parameter (Scheme 2) in the Hanzhong basin.

| | NSE | LNSE | RMSE_Q5 | RMSE_Q20 | RMSE_mid | RMSE_Q70 | RMSE_Q95 |
|---|---|---|---|---|---|---|---|
| Calibration | | | | | | | |
| Scheme 1 | 0.379 | 0.664 | 0.524 | 0.146 | 0.2 | 0.13 | 0.426 |
| Scheme 2 | 0.383 | 0.672 | 0.517 | 0.164 | 0.18 | 0.127 | 0.425 |
| Verification | | | | | | | |
| Scheme 1 | 0.152 | 0.514 | 0.511 | 0.203 | 0.4 | 0.152 | 0.598 |
| Scheme 2 | 0.149 | 0.542 | 0.493 | 0.24 | 0.246 | 0.138 | 0.635 |
| Verification-calibration | | | | | | | |
| Scheme 1 | 0.773 | 0.85 | -0.013 | 0.057 | 0.2 | 0.022 | 0.172 |
| Scheme 2 | 0.766 | 0.87 | -0.024 | 0.076 | 0.066 | 0.011 | 0.21 |

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
