# Peer review of "A framework for seasonal variations of hydrological model parameters: Impact on model results and response to dynamic catchment characteristics"

_Hydrology and Earth System Sciences, 2020_

## Referee Comment (RC1) · Anonymous Referee #1 · 1 Jul 2020

General

This study touches upon two very important but distinct topics in hydrological modelling: (1) temporal variation of model parameters and (2) model calibration issues: finding optima in a high-dimensional parameter space with a potentially rugged objective function landscape. The study finds that dynamic parameters increase the performance of the HYMOD model in validation setting, but have a poor correspondence to observed dynamic catchment characteristics. Even though the study addresses two very relevant topics, it remains unclear what the general value of the findings in this study are. The authors need to demonstrate this more clearly before I can recommend publication of

this manuscript.

I have some fundamental objections to the approach of "dynamic parameters" that the authors use. The term "dynamic parameters" gives the impression that the parameters vary in time, i.e. during model simulations. As I understand it, this is not the case. The model is just fitted to different time periods individually with static parameters. Due to the considerable temporal memory in the model states, the parameter values fitted to a specific sub-period will, however, also be influenced by the period(s) before, not only the period it is intended to represent. It is unclear to which degree this is the case and to which degree it influences the results.

Pg19, Line 25-26: It is unclear what the methods were exactly that led to this conclusion. When looking at Table A1, how is the dynamic case calculated? I am missing some equations here. Furthermore, I see that the model performance is very poor for that catchment, independent of the "static" and the "dynamic" approach. How is the model performance for the other catchments? NSE-values of around 0.15 in the validation period are rather worrying. How do you explain that?

Conclusions section

Pg19 Line 26-27: This conclusion is not supported by the results. It is not shown that correlations between parameters is the reason for the missing improvement achieved with a single dynamic parameter.

Pg20 Line 5-11: The meaning of this paragraph is especially elusive, and it is unclear how these statements refer to the analysis presented in the manuscript. In general, the conclusion section is not very clear and it also doesn't clarify in which way this study is of general value.

Abstract

Line 11: remove "however". "received little attention" is not true if it refers to constant parameters. If it refers to dynamic parameters this is true, but then repeat "dynamic" to

make it clear.

Line 15: "probability distributions of violin plots" is an awkward formulation. "probability distributions visualized in violin plots" would be more accurate.

Line 21 "response [. . .] is generally poor": not clear what this means. Poor in what sense?

Text

Pg2 Line 4: "for dynamic" and static "parameters in hydrological" would be more accurate.

Pg2 Line 14: did you mean: "the more local optima there are"?

Pg2 Line 32: How does this sentence relate to the previous part of the paragraph? The meaning of the sentence is not clear. What does "measurement of water resources problems" mean?

Pg7 Line 15-22: The authors suggest that the width of the parameter distribution is only a function of the number of local minima. This is not true, wide (or "flat") distributions can also be unimodal, in that case, the parameters are just relatively insensitive to the objective function. This needs to be clarified throughout the manuscript (e.g. also on Pg 12, Line 12-13)

Pg9 Line 17: The authors repeatedly speak of "the divergence measure". This sounds like there is a quantitative measure to describe the "divergence", which I assume is a measure of the width of a distribution. The term "divergence" is, however, confusing since it has been commonly used for the difference between two distributions, which here is not the case. I suggest the authors replace "divergence" with "width of the parameter distribution" or "standard deviation". Also, I would not call it "measure" unless it is a quantitative indicator.

Pg12 Line 5: I don't understand what the authors want to say with this sentence. Figure

7 is referred as Figure 8 in the manuscript.

Pg16 Line 17-18: It is not clear what the methods are that were used for this analysis. Please describe in more detail and move it to the methods section if necessary.

Table A1: the lower part ("verification-calibration", I think this is supposed to be the difference between validation and calibration) does not add up for columns "NSE" and "LNSE"

[Figure]

---

## Referee Comment (RC2) · 6 Aug 2020

General comments The manuscript identifies the problem of finding the global optimum for dynamic hydrological model parameters and proposes an approach involving the investigation of their evolutionary processes. The study was performed for data from three river basins: Hanzhong, Mumahe and Xunhe. The Authors used hydrological and climatic data from the period 1980-1990. Two clustering operations have been performed on this data. Additionally, both data groups were divided into 4 sub-periods: dry period and three wet periods. The data were analyzed using maximal information coefficient (MIC) and the Principal Component Analysis (PCA). The HYMOD model was

used. The parameters used in this model were analyzed in the paper (5 parameters). The model has been calibrated. Moreover, the Authors used the Shuffled Complex Evolution algorithm from the University of Arizona (SCE-UA) as an evolutionary algorithm for dynamic parameters. The combination of the Nash-Sutcliffe Efficiency index (NSE) and its logarithmic transformation (LNSE) was used as the function of the object. The simulation performance with dynamic parameters was assessed using seven performance metrics including NSE, LNSE, a five-segment flow duration curve (5FDC) with the Root Mean Square Error (RMSE). A fitness landscape was used to visualize the evolutionary processes, and violin plots were used to visualize the distribution parameters. The Authors collected a large number of results that are presented in the charts. These charts are clear and interesting. Such studies are undoubtedly needed because finding the global optimum for dynamic hydrological model parameters is an important practical issue. In my opinion, the novelty of this work is in the developed framework for the dynamic operation of parameters. The Authors might also consider expanding the discussion to the case of single- and multi-parameter spaces. In my opinion, the supplement (Supporting Information), could contain: equations, codes, details of used parameters or the names of used programs. I am interested in how the data were prepared for the determination of distributions and for the MIC and PCA analyzes. Were the data logged in the PCA analysis?

I have a few more questions / suggestions, which I include as specific comments: Line 17, page 2: The concept of an evolutionary process described in the introduction is not very clear. Please consider a more detailed description. Line 22, page 3: Please consider using "hydrological and climatic data" instead of "daily streamflow and climatic data". Have the authors considered including water temperature and air temperature in the analyzes? Line 6, page 4: I suggest that the methodology for performing PCA and MIC analyzes should be described. Line 9, page 4: What do the Authors understand by total precipitation? Is it the annual rainfall? Line 4, page 5: if the code is open, please consider making it available in a supplement. Line 12, page 5: A description or explanation of the 5 parameters mentioned would be desirable. [reference to the

supplement] Line 2, page 20: The use of the CDF (cumulative distribution function) is mentioned. If these results are not presented in the article, I suggest that they should not appear in the conclusions.

---

## Author Comment (AC1) · 9 Sep 2020

**Replies to Referee #1**

**Title: Dynamics of hydrological model parameters: mechanisms, problems, and solution**
**General comments:**
This study touches upon two very important but distinct topics in hydrological modelling:
(1) temporal variation of model parameters and (2) model calibration issues: finding optima in a high-dimensional parameter space with a potentially rugged objective function landscape. The study finds that dynamic parameters increase the performance of the HYMOD model in validation setting, but have a poor correspondence to observed dynamic catchment characteristics. Even though the study addresses two very relevant topics, it remains unclear what the general value of the findings in this study are. The authors need to demonstrate this more clearly before I can recommend publication of this manuscript.

**Reply:** We thank the reviewer for the positive evaluation and constructive comments. We agree with the reviewer that we need to improve the presentation of the manuscript by stating better the overall value of the findings of the study, and by presenting better the relation between the two topics in the hydrological model, i.e. (1) temporal variation of model parameters and (2) model calibration issues.

By doing this, in the revised version, we will improve the statements of the motivation and novelty of the study, the objective, the study procedure, and the main findings as follows. Previous studies have shown that the seasonal dynamics of model parameters can compensate for structural defects of hydrological models and improve the accuracy of the streamflow forecast to some extent. However, some fundamental issues for improving model performance with seasonal dynamic parameters still need to be addressed. The aim of the study is dedicated to 1) proposing a novel framework for seasonal variations of hydrological model parameters to improve the model performance, 2) expanding the discussion on the response of seasonal dynamic parameters to dynamic catchment characteristics. The procedure of the framework is developed with (1) extraction of the dynamic catchment characteristics using current data mining techniques, (2) sub-period calibration operations for seasonal dynamic parameters, considering the effects of the significant correlation between the parameters, the number of multiplying parameters, and the temporal memory in the model states in two adjacent sub-periods on calibration operations, and (3) multi-metric assessment of model performance designed for various flow phases. The main findings are 1) the proposed framework significantly improved the accuracy and robustness of the model, 2) however, there was generally poor response of seasonal dynamic parameter set to catchment dynamics. Namely, the dynamic changes of parameters did not follow the dynamics of catchment characteristics. Hence, we deepen the discussion on the poor response in terms of (1) the evolutionary processes on seasonal dynamic parameters optimized by global optimization, considering that the possible failure in finding the global optimum might lead to unreasonably seasonal dynamic parameter values. Moreover, a practical tool for visualizing the evolutionary processes of seasonal dynamic parameters was designed using geometry visualization techniques. (2) The strong correlation between parameters, considering the dynamic changes of one parameter might be intervened by other parameters due to their interdependence. Consequently, the poor response of seasonal dynamic parameter set to dynamic catchment characteristics may be attributed in part to the possible failure in finding the global optimum and strong correlation between parameters. Further analysis also revealed that even though individual parameters

cannot respond well to dynamic catchment characteristics, and a dynamic parameter set could carry the information extracted from dynamic catchment characteristics and improve the model performance.

In addition, in order to take full consideration of the reviewer's advice and achieve the above improvement, we will slightly modify the title of the manuscript as: "Developing a framework for seasonal variations of hydrological model parameters: Expanding the discussion on the response of seasonal dynamic parameters to dynamic catchment characteristics".

I have some fundamental objections to the approach of "dynamic parameters" that the authors use. The term "dynamic parameters" gives the impression that the parameters vary in time, i.e. during model simulations. As I understand it, this is not the case. The model is just fitted to different time periods individually with static parameters. Due to the considerable temporal memory in the model states, the parameter values fitted to a specific sub-period will, however, also be influenced by the period(s) before, not only the period it is intended to represent. It is unclear to which degree this is the case and to which degree it influences the results.

**Reply:** Thanks for the comment. Our response to the comment is as follows. We first explain our original consideration, and then how we will make changes according to the reviewer's suggestion. First, we explain the original consideration and the procedure that followed. For "dynamic parameters", the proposed framework in this study first extracted the dynamic catchment characteristics using a series of data mining methods. Then, hydrological processes clustering, as a bridge, was built between extracted information of dynamic catchment characteristics and calibration operations of the hydrological model. The calibration operation for seasonal dynamic parameters is also called sub-period calibration. The static parameters are seasonally dynamized. Even though more techniques for dynamics of hydrological model parameters have been developed, such as parameters vary in time during model simulations (Xiong et al., 2019; Motavita et al., 2019; Manfreda et al., 2018; Lan et al., 2018, 2020; Fowler et al., 2018), the proposed sub-period calibration effectively integrated into data mining techniques to compensate for the structural defects of the traditional hydrological models with static parameters. The framework could fully utilize the extracted information of dynamic catchment characteristics and improve model performance. Second, we agree with the reviewer that this kind of seasonal variation of parameters is not a full dynamic parameter set, and to better distinguish the concept of dynamic parameters that change continuously with time or space, this study will change "dynamic parameters" to "seasonal dynamic parameters" or "seasonal variations of parameters". All related explanations will be clarified in the revised manuscript.

For considerable temporal memory in the model states in adjacent sub-periods, a controlled trial will be added in the revised manuscript. It will be used to explore that how the fluxes and state variables in a certain sub-period were affected by the previous period, due to the considerable temporal memory in the model states while shifting of the parameter set between two adjacent sub-periods. Motivated by this comment from the Referee, five calibration operations would be added and compared to analyze the effects of the significant correlation between the parameters, the number of multiplying parameters, and the temporal memory in the model states on the sub-period calibration. The specific information for five calibration operations is as follows (we have completed this work):

[revised manuscript text omitted]

Pg19, Line 25-26: It is unclear what the methods were exactly that led to this conclusion. When looking at Table A1, how is the dynamic case calculated? I am missing some equations here. Furthermore, I see that the model performance is very poor for that catchment, independent of the "static" and the "dynamic" approach. How is the model performance for the other catchments? NSE-values of around 0.15 in the validation period are rather worrying. How do you explain that?

**Reply:** We agree with the comment of the Referee. The inappropriate or unclear descriptions will be removed in Pg19, Line 25-26. For the issue that the effects of the significant correlation between the parameters on the proposed framework, calibration operation **II** will be designed in the revised manuscript (see the reply to the previous question). Compared with the operation **I** with static parameters, the seasonal dynamics of a single parameter $K_s$ with high sensitivity in operation **II** (see Figure 4a) do not significantly improve or decrease model performance. The result is consistent with Bárdossy (2007). The author demonstrated that one dynamic parameter might be compensated for the adjustment of other time-invariant parameters while modeling calibration due to the strong correlation between parameters. As a result, the final performance of the model is not significantly improved. Figure 4b verifies that there is a significant linear and nonlinear correlation between parameters by MIC coefficients.

Sorry, this was a mistake in Table A1. The model performance will be presented in Figure 3b (see the reply to the previous question). Also, details about the method including program codes will be opened and attached in Supporting Information.

**Conclusions section:**

Pg19 Line 26-27: This conclusion is not supported by the results. It is not shown that correlations between parameters is the reason for the missing improvement achieved with a single dynamic parameter.

**Reply:** Thanks. The experiment is redesigned. The reply is provided in the previous question.

Pg20 Line 5-11: The meaning of this paragraph is especially elusive, and it is unclear how these statements refer to the analysis presented in the manuscript. In general, the conclusion section is not very clear and it also doesn't clarify in which way this study is of general value.

**Reply:** Sorry that the conclusion was not clear enough. The conclusions will be rewritten in the revised manuscript. The specific information is as follows:

"**6 Conclusions**

The seasonal dynamic of parameters is one of the practical approaches for compensating structural defects of hydrological models and improving model performance. In this study, a framework was proposed to extract the dynamic catchment characteristics using a series of data mining methods. The information extraction included selection and generation of climate and land-use indices, screening of indices, processing of redundant information among indices and clustering of hydrological processes based on the indices. The extracted information and model calibration were effectively integrated by sub-period calibration operations. The recommended calibration operation considered the sensitivity and correlation of parameters, the dimensions of parameters, and considerable temporal memory in the model states between two adjacent subperiods. Multi-metric assessment of model performance was designed for various flow phases and the temporal transitivity of parameters.

The study showed that the proposed framework significantly improves the accuracy and robustness of the hydrological model. However, there was generally a poor response of seasonal dynamic parameter set to dynamic catchment characteristics. Hence, the investigation for this issue was expanded considering the evolutionary processes on seasonal dynamic parameters optimized by global optimization and the intricate and significant correlation between parameters. Consequently, the poor response of seasonal dynamic parameter set to catchment dynamics might be attributed in part to the possible failure in finding the global optimum when optimizing the seasonal dynamic parameters and strong correlation between parameters. Even though individual parameters could not respond well to dynamic catchment characteristics, a dynamic parameter set could carry the information extracted from dynamic catchment characteristics and improve the model performance. In addition, a novel tool for visualizing the evolutionary processes of seasonal dynamic parameters was designed using geometry visualization techniques, which was also regarded as an important tool to understand the model running with dynamic hydrological model parameters in the next research. More case studies and applications of hydrological models can be performed in the future. They are expected to yield insights into the predictive performance of hydrological models."

**Abstract section:**

Line 11: remove "however". "received little attention" is not true if it refers to constant parameters. If it refers to dynamic parameters this is true, but then repeat "dynamic" to make it clear.

**Reply:** Thanks for the referee's reminding. The revision will be completed, as suggested.

Line 15: "probability distributions of violin plots" is an awkward formulation. "probability distributions visualized in violin plots" would be more accurate.
**Reply:** Thanks for the advice. The revision will be done, as suggested.

Line 21 "response [. . .] is generally poor": not clear what this means. Poor in what sense?
**Reply:** The explanation will be added in the Abstract section, as follows.
"There is generally poor response of seasonal dynamic parameter set to dynamic catchment characteristics. Namely, the dynamic changes of parameters did not follow the dynamics of catchment characteristics."

**Text section:**
Pg2 Line 4: "for dynamic" and static "parameters in hydrological" would be more accurate.
**Reply:** Thanks, it will be corrected.

Pg2 Line 14: did you mean: "the more local optima there are"?
**Reply:** Yes, revision will be made, as suggested.

Pg2 Line 32: How does this sentence relate to the previous part of the paragraph? The meaning of the sentence is not clear. What does "measurement of water resources problems" mean?
**Reply:** Thanks for the comment. Ambiguous sentence will be rewritten as follows. "However, the simple and practical method for hydrological modeling with seasonal dynamic parameters still needs to be further explored."

Pg7 Line 15-22: The authors suggest that the width of the parameter distribution is only a function of the number of local minima. This is not true, wide (or "flat") distributions can also be unimodal, in that case, the parameters are just relatively insensitive to the objective function. This needs to be clarified throughout the manuscript (e.g. also on Pg 12, Line 12-13).
**Reply:** Thanks, we will clarify that the probability distributions in violin plots are used to visualize the evolutionary processes by characterizing the structures of fitness landscapes in this study. With an adequate parameter space, namely the same number of iterations, wide (or "flat") distributions might not be relatively unimodal compared with other parameters in a hydrological model.

Pg9 Line 17: The authors repeatedly speak of "the divergence measure". This sounds like there is a quantitative measure to describe the "divergence", which I assume is a measure of the width of a distribution. The term "divergence" is, however, confusing since it has been commonly used for the difference between two distributions, which here is not the case. I suggest the authors replace "divergence" with "width of the parameter distribution" or "standard deviation". Also, I would not call it "measure" unless it is a quantitative indicator.
**Reply:** Thanks for the Referee's comment and advice. The "the divergence measure" will be replaced with "the width of the parameter set distribution", as suggested.

Pg12 Line 5: I don't understand what the authors want to say with this sentence. Figure 7 is referred as Figure 8 in the manuscript.

**Reply:** The ambiguous statement will be changed to "The parameter $K_s$ presents the thinner distribution of violin plots in all sub-periods". Figure 7 will be revised to Figure 8.

Pg16 Line 17-18: It is not clear what the methods are that were used for this analysis. Please describe in more detail and move it to the methods section if necessary.

**Reply:** Thanks. The experiment (operation II) is redesigned. The reply is provided in the previous question.

Table A1: the lower part ("verification-calibration", I think this is supposed to be the difference between validation and calibration) does not add up for columns "NSE" and "LNSE".

**Reply:** Sorry, this was a mistake in Table A1 that will be removed. The model performance will be presented in Figure 3b (see the reply to the previous question).

---

## Author Response (AR1)

**Replies to Gruss, Łukasz**

**Title: Dynamics of hydrological model parameters: mechanisms, problems, and solution**
**General comments:**
The manuscript identifies the problem of finding the global optimum for dynamic hydrological model parameters and proposes an approach involving the investigation of their evolutionary processes. The study was performed for data from three river basins: Hanzhong, Mumahe and Xunhe. The Authors used hydrological and climatic data from the period 1980-1990. Two clustering operations have been performed on this data. Additionally, both data groups were divided into 4 sub-periods: dry period and three wet periods. The data were analyzed using maximal information coefficient (MIC) and the Principal Component Analysis (PCA). The HYMOD model was used. The parameters used in this model were analyzed in the paper (5 parameters). The model has been calibrated. Moreover, the Authors used the Shuffled Complex Evolution algorithm from the University of Arizona (SCE-UA) as an evolutionary algorithm for dynamic parameters. The combination of the Nash-Sutcliffe Efficiency index (NSE) and its logarithmic transformation (LNSE) was used as the function of the object. The simulation performance with dynamic parameters was assessed using seven performance metrics including NSE, LNSE, a five-segment flow duration curve (5FDC) with the Root Mean Square Error (RMSE). A fitness landscape was used to visualize the evolutionary processes, and violin plots were used to visualize the distribution parameters. The Authors collected a large number of results that are presented in the charts. These charts are clear and interesting. Such studies are undoubtedly needed because finding the global optimum for dynamic hydrological model parameters is an important practical issue. In my opinion, the novelty of this work is in the developed framework for the dynamic operation of parameters. The Authors might also consider expanding the discussion to the case of single- and multi-parameter spaces. In my opinion, the supplement (Supporting Information), could contain: equations, codes, details of used parameters or the names of used programs. I am interested in how the data were prepared for the determination of distributions and for the MIC and PCA analyzes. Were the data logged in the PCA analysis?

**Reply:** We thank Dr. Łukasz Gruss for reviewing our paper and for your positive evaluation and encouragement. All your comments and suggestions have been fully considered in making revisions. All codes of methods and data in this study have been opened and attached in Supporting Information. Moreover, we have carefully studied, considered and responded to all comments point-by-point as follows. For clarity, all comments are given in black and responses are given in the blue text. The revised parts in our manuscript are highlighted in red.

**Specific comments:**
Line 17, page 2: The concept of an evolutionary process described in the introduction is not very clear. Please consider a more detailed description.
**Reply:** Thanks for the referee's reminding. More detailed descriptions have been added in the Introduction section. The specific information is as follows:
  "Evolutionary algorithms (EAs) are the most well-established class of global optimization algorithms for solving water resources problems (Maier et al., 2014). In each evolutionary process, four steps, including evaluation, fitness assignment, selection and

reproduction, are performed. The parameter set with the best objective function value in each evolutionary process loop is recorded in the "evolutionary processes". The evolutionary process evolves toward minimizing the objective function values. The final optimum is obtained at the end of the run while satisfying the stopping criteria."

Line 22, page 3: Please consider using "hydrological and climatic data" instead of "daily streamflow and climatic data". Have the authors considered including water temperature and air temperature in the analyzes?

**Reply:** Thanks for the Referee's suggestion. Revision has been completed, as suggested. We did not use water temperature and air temperature in the analyzes.

Line 6, page 4: I suggest that the methodology for performing PCA and MIC analyzes should be described.

**Reply:** We agree with the Referee on this point. The methodology for performing PCA and MIC analyzes has been described in the revised manuscript for the sake of readability. The detailed information is as follows:

"A set of climatic-land surface indices was provided and preprocessed using the maximal information coefficient (MIC) and principal components analysis (PCA). Actually, the indices are specified based on dynamic characteristics on a catchment. The climate and land-surface indices were selected just as examples in this study. The selected climatic indices included total precipitation, maximum 1-day precipitation, maximum five-day precipitation, moderate precipitation days, heavy precipitation days, total pan evaporation, maximum 1-day pan evaporation and minimum 1-day pan evaporation. The land-surface indices included antecedent streamflow and runoff coefficient. The definition of the indices is provided in Table A1. Indeed, the indices that are independent with streamflow may damage the extraction of dynamic catchment characteristics. Hence, the selected indices should be screened first by identifying the degree of correlation between the indices and streamflow. The MIC, as a statistical metric, can indicate the linear and nonlinear correlation between the variables (Zhang et al., 2014) and is used to screen the indices in this study. The detailed introduction of the MIC metric is provided in the Supporting Information. It is assumed that the indices have a significant effect on streamflow and are picked up while the MIC value is larger than 0.35. In addition, a large amount of redundant information still exists among the screened indices and damages the availability of the extracted information. Hence, PCA is applied to further eliminate the multicollinearity of indices (Ho et al., 2017)."

Line 9, page 4: What do the Authors understand by total precipitation? Is it the annual rainfall?

**Reply:** The definitions of climatic-land surface indices have been supplemented in the Appendix as follows:

**Table A1.** Climatic-land surface indices.

| Indices | Descriptive names | Definitions | Units |
|---------|-------------------|-------------|-------|
| $R_T$ | Total precipitation | Current half-monthly total precipitation | mm |
| RX1day | Maximum 1-day precipitation | Half-monthly highest 1-day precipitation | mm |

| | | | |
|---|---|---|---|
| RX5day | Maximum five-day precipitation | Half-monthly highest consecutive 5-day precipitation | mm |
| R25pday | Moderate precipitation days | Count of days where RR (daily precipitation amount) < 25th percentile | days |
| R75pday | Heavy precipitation days | Count of days where RR ≥75th percentile | days |
| $PE_T$ | Total pan evaporation | Current half-monthly total pan evaporation | mm |
| $PE_x$ | Maximum 1-day pan evaporation | Half-monthly highest 1-day pan evaporation | mm |
| $PE_n$ | Minimum 1-day pan evaporation | Half-monthly lowest 1-day pan evaporation | mm |
| $Q_{T-1}$ | Antecedent streamflow | Antecedent half-monthly average streamflow | $m^3/s$ |
| $C$ | Runoff coefficient | Ratio of runoff volume to rainfall volume | |

Line 4, page 5: if the code is open, please consider making it available in a supplement.

**Reply:** All codes of methods and data have been opened and attached in Supporting Information of the revised manuscript.

Line 12, page 5: A description or explanation of the 5 parameters mentioned would be desirable. [reference to the supplement]

**Reply:** The definitions of parameters, state variables and fluxes used in the HYMOD model have been supplemented in the Appendix as follows:

**Table A2.** Definitions of parameters, state variables and fluxes used in the HYMOD model (Wagener et al., 2001).

| Label | Property | Range | Description |
|---|---|---|---|
| $H_{uz}$ | Parameter | 0-1000 [mm] | Maximum height of soil moisture accounting tank |
| $B$ | Parameter | 0-1.99 | Scaled distribution function shape |
| alpha | Parameter | 0-0.99 | Quick/slow split |
| $K_q$ | Parameter | 0-0.99 | Quick-flow routing tanks' rate |
| $K_s$ | Parameter | 0-0.99 | Slow-flow routing tank's rate |
| $XH_{uz}$ | State variable | [mm] | Upper zone soil moisture tank state height |
| $XC_{uz}$ | State variable | [mm] | Upper zone soil moisture tank state contents |
| $X_q$ | State variable | [mm] | Quick-flow tank states contents |
| $X_s$ | State variable | [mm] | Slow-flow tank state contents |
| $AE$ | Fluxes | [mm/day] | Actual evapotranspiration flux |
| $OV$ | Fluxes | [mm/day] | Precipitation excess flux |
| $Q_q$ | Fluxes | [mm/day] | Quick-flow flux |
| $Q_s$ | Fluxes | [mm/day] | Slow-flow flux |
| $Q_{sim}$ | Fluxes | [mm/day] | Total streamflow flux |

Line 2, page 20: The use of the CDF (cumulative distribution function) is mentioned. If these results are not presented in the article, I suggest that they should not appear in the conclusions.

**Reply:** The use of the CDF has been removed, as suggested.

**References:**

Gomez, J.: Stochastic global optimization algorithms: A systematic formal approach, Information Sciences, 472, 53-76, https://doi.org/10.1016/j.ins.2018.09.021, 2019.

Ho, M., Lall, U., Sun, X., and Cook, E. R.: Multiscale temporal variability and regional patterns in 555 years of conterminous U.S. streamflow, 53, 3047-3066, 10.1002/2016wr019632, 2017.

Maier, H. R., Kapelan, Z., Kasprzyk, J., Kollat, J., Matott, L. S., Cunha, M. C., Dandy, G. C., Gibbs, M. S., Keedwell, E., Marchi, A., Ostfeld, A., Savic, D., Solomatine, D. P., Vrugt, J. A., Zecchin, A. C., Minsker, B. S., Barbour, E. J., Kuczera, G., Pasha, F., Castelletti, A., Giuliani, M., and Reed, P. M.: Evolutionary algorithms and other metaheuristics in water resources: Current status, research challenges and future directions, Environ Modell Softw, 62, 271-299, 10.1016/j.envsoft.2014.09.013, 2014.

Wagener, T., Boyle, D. P., Lees, M. J., Wheater, H. S., Gupta, H. V., and Sorooshian, S.: A framework for development and application of hydrological models, Hydrol. Earth Syst. Sci., 5, 13-26, 10.5194/hess-5-13-2001, 2001.

Zhang, Y., Jia, S., Huang, H., Qiu, J., and Zhou, C.: A novel algorithm for the precise calculation of the maximal information coefficient, Sci Rep, 4, 6662, 10.1038/srep06662, 2014.

**Replies to Referee #1**

**Title: Dynamics of hydrological model parameters: mechanisms, problems, and solution**
**General comments:**
This study touches upon two very important but distinct topics in hydrological modelling:
(1) temporal variation of model parameters and (2) model calibration issues: finding optima in a high-dimensional parameter space with a potentially rugged objective function landscape. The study finds that dynamic parameters increase the performance of the HYMOD model in validation setting, but have a poor correspondence to observed dynamic catchment characteristics. Even though the study addresses two very relevant topics, it remains unclear what the general value of the findings in this study are. The authors need to demonstrate this more clearly before I can recommend publication of this manuscript.

**Reply:** We thank the reviewer for the positive evaluation and constructive comments. We agree with the reviewer that we need to improve the presentation of the manuscript by stating better the overall value of the findings of the study, and by presenting better the relation between the two topics in the hydrological model, i.e. (1) temporal variation of model parameters and (2) model calibration issues. Moreover, we have carefully studied, considered and responded to all comments point-by-point as follows. For clarity, all comments are given in black and responses are given in the blue text. The revised parts in our manuscript are highlighted in red.

Following the reviewer's advice and suggestion, in the revised version, we have improved the statements of the motivation and novelty of the study, the objective, the study procedure, and the main findings as follows. Previous studies have shown that the seasonal dynamics of model parameters can compensate for structural defects of hydrological models and improve the accuracy of the streamflow simulations to some extent. However, some fundamental issues for improving model performance with seasonal dynamic parameters still need to be addressed.

The aim of the study is therefore dedicated to 1) proposing a novel framework for seasonal variations of hydrological model parameters to improve the model performance, and 2) expanding the discussion on impact on model results and response of seasonal dynamic parameters to dynamic catchment characteristics.

The procedure of the framework is developed with (1) extraction of the dynamic catchment characteristics using current data mining techniques, (2) sub-period calibration operations for seasonal dynamic parameters, considering the effects of the significant correlation between the parameters, the number of multiplying parameters, and the temporal memory in the model states in two adjacent sub-periods on calibration operations, and (3) multi-metric assessment of model performance designed for various flow phases.

The main findings are 1) the proposed framework significantly improved the accuracy and robustness of the model, 2) however, there was generally poor response of seasonal dynamic parameter set to catchment dynamics. Namely, the dynamic changes of parameters did not follow the dynamics of catchment characteristics. Hence, we deepen the discussion on the poor response in terms of (1) the evolutionary processes on seasonal dynamic parameters optimized by global optimization, considering that the possible failure in finding the global optimum might lead to unreasonable seasonal dynamic parameter values. Moreover, a practical tool for visualizing the evolutionary processes of seasonal dynamic parameters was designed using geometry visualization techniques. (2) The strong correlation between parameters, considering the dynamic changes of one parameter might be intervened by other parameters

due to their interdependence. Consequently, the poor response of seasonal dynamic parameter set to dynamic catchment characteristics may be attributed in part to the possible failure in finding the global optimum and strong correlation between parameters. Further analysis also revealed that even though individual parameters cannot respond well to dynamic catchment characteristics, and a seasonal dynamic parameter set could carry the information extracted from dynamic catchment characteristics and improve the model performance.

In addition, in order to take full consideration of the reviewer's advice and achieve the above improvement, we have slightly modified the title of the manuscript as: "A framework for seasonal variations of hydrological model parameters: Impact on model results and response to dynamic catchment characteristics".

I have some fundamental objections to the approach of "dynamic parameters" that the authors use. The term "dynamic parameters" gives the impression that the parameters vary in time, i.e. during model simulations. As I understand it, this is not the case. The model is just fitted to different time periods individually with static parameters. Due to the considerable temporal memory in the model states, the parameter values fitted to a specific sub-period will, however, also be influenced by the period(s) before, not only the period it is intended to represent. It is unclear to which degree this is the case and to which degree it influences the results.

**Reply:** Thanks for the comment. This comment covers two issues. One is the use of the team "dynamic parameters" in the manuscript, and another is the possible influence of temporal memory of the catchment. Our response to the comment is as follows. We first explain our original consideration, and then how we made changes according to the reviewer's suggestion. First, for "dynamic parameters", the proposed framework in this study first extracted the dynamic catchment characteristics using a series of data mining methods. Then, hydrological processes clustering, as a bridge, was built between extracted information of dynamic catchment characteristics and calibration operations of the hydrological model. The calibration operation is also called sub-period calibration. The static parameters are seasonally dynamized. The proposed sub-period calibration effectively integrated into data mining techniques to compensate for the structural defects of the traditional hydrological models with static parameters. The framework fully utilized the extracted information on dynamic catchment characteristics and improve model performance. Second, we agree with the reviewer that this kind of seasonal variation of parameters is not a full dynamic parameter set, and to better distinguish the concept of dynamic parameters that change continuously with time or space, this study changed "dynamic parameters" to "seasonal dynamic parameters" or "seasonal variations of parameters". All related explanations have been clarified in the revised manuscript.

To demonstrate and quantify the influence of temporal memory in the model states in adjacent sub-periods, a controlled trial was added in the revised manuscript. It was used to explore that how the fluxes and state variables in a certain sub-period were affected by the previous period, due to the temporal memory in the model states and correlation between parameters. Five calibration operations have been added and compared to analyze the effects of the significant correlation between the parameters, the number of multiplying parameters, and the temporal memory in the model states on the sub-period calibration. The specific information for five calibration operations is as follows:

[revised manuscript text omitted]

Pg19, Line 25-26: It is unclear what the methods were exactly that led to this conclusion. When looking at Table A1, how is the dynamic case calculated? I am missing some equations here. Furthermore, I see that the model performance is very poor for that catchment, independent of the "static" and the "dynamic" approach. How is the model performance for the other catchments? NSE-values of around 0.15 in the validation period are rather worrying. How do you explain that?

**Reply:** We agree with the comment of the Referee. The inappropriate or unclear descriptions have been removed in Pg19, Line 25-26. For the issue that the effects of the significant correlation between the parameters on the proposed framework, calibration operation **II** has been designed in the revised manuscript (see the reply to the previous question). Compared with the operation **I** with static parameters, the seasonal dynamics of a single parameter $K_s$ with high sensitivity in operation **II** (see Figure 4a) do not significantly improve or decrease model performance. The result is consistent with Bárdossy (2007). The author demonstrated that one dynamic parameter might be compensated for the adjustment of other time-invariant parameters while modeling calibration due to the strong correlation between parameters. As a result, the final performance of the model is not significantly improved. Figure 4b verifies that there is a significant linear and nonlinear correlation between parameters by MIC coefficients.

Sorry, this was a mistake in Table A1. The model performance has been presented in Figure 3b (see the reply to the previous question). Also, details about the method including program codes were opened and attached in Supporting Information.

**Conclusions section:**

Pg19 Line 26-27: This conclusion is not supported by the results. It is not shown that correlations between parameters is the reason for the missing improvement achieved with a single dynamic parameter.

**Reply:** Thanks. The experiment was redesigned, which demonstrates the influence of correlations between parameters on model results. The reply is provided in the revised manuscript. The specific information is as follows:

In the introduction section, the question for correlations between seasonal dynamic parameters was raised. (1) How does the potential correlation between parameters affect the sub-period calibration? Can the seasonal dynamic of a single parameter with high sensitivity or identification effectively improve the simulation performance of hydrological models without considering the correlation between parameters?

In the methods section, the operation was designed. In operation II, the linear and nonlinear correlation between parameters is first investigated using MIC. Then, a simple but useful tool, i.e., a scatter plot (Paruolo et al., 2013), is used for identifying the sensitive parameter of hydrological models. Only the sensitive parameter is considered as of potential seasonal dynamic parameter, but other parameters are time-invariant.

In the results section, the results were provided. Compared with the operation **I** (controlled trial), the seasonal dynamics of a single parameter $K_s$ with high sensitivity (see Figure 4a) do not significantly improve or decrease model performance in operation **II**. The result is consistent with Bárdossy (2007). The author demonstrated that one dynamic parameter might be compensated for the adjustment of other time-invariant parameters during calibration due to the strong correlation between parameters. As a result, the final performance of the model with the single dynamic parameter is not significantly improved. Figure 4b verifies that there is a significant linear and nonlinear correlation between parameters by MIC coefficients.

In the discussion section, the results were elaborated. According to the values of MIC shown in Figure 4b, the significant linear and nonlinear correlation MIC existed between parameters, which verify the Bárdossy's (2007) view. Namely, the dynamic changes of one parameter might be intervened by other parameters due to their interdependence. However, according to the assessment results of model performance, the model performance with a seasonal dynamic parameter set has significant improvement. Even though individual parameters cannot respond well to dynamic catchment characteristics, a dynamic parameter set could carry the information extracted from dynamic catchment characteristics and improve the model performance.

Pg20 Line 5-11: The meaning of this paragraph is especially elusive, and it is unclear how these statements refer to the analysis presented in the manuscript. In general, the conclusion section is not very clear and it also doesn't clarify in which way this study is of general value.

**Reply:** Sorry that the conclusion was not clear enough. The conclusions have been rewritten in the revised manuscript. The specific information is as follows:

"**6 Conclusions**

The seasonal dynamic of parameters is one of the practical approaches for compensating structural defects of hydrological models and improving model performance. In this study, a framework was proposed to extract the dynamic catchment characteristics using a series of data

mining methods. The information extraction included selection and generation of climate and land-use indices, screening of indices, processing of redundant information among indices and clustering of hydrological processes based on the indices. The extracted information and model calibration were effectively integrated by sub-period calibration operations. The recommended calibration operation considered the sensitivity and correlation of parameters, the dimensions of parameters, and temporal memory in the model states between two adjacent subperiods. Multi-metric assessment of model performance was designed for various flow phases and the temporal transitivity of parameters.

The study showed that the proposed framework significantly improves the accuracy and robustness of the hydrological model. However, there was generally a poor response of seasonal dynamic parameter set to dynamic catchment characteristics. Hence, the investigation for this issue was expanded considering the evolutionary processes on seasonal dynamic parameters optimized by global optimization and the intricate and significant correlation between parameters. Consequently, the poor response of seasonal dynamic parameter set to catchment dynamics might be attributed in part to the possible failure in finding the global optimum when optimizing the seasonal dynamic parameters and strong correlation between parameters. Even though individual parameters could not respond well to dynamic catchment characteristics, a seasonal dynamic parameter set could carry the information extracted from dynamic catchment characteristics and improve the model performance. In addition, a novel tool for visualizing the evolutionary processes of seasonal dynamic parameters was designed using geometry visualization techniques, which was also regarded as an important tool to understand the model running with dynamic hydrological model parameters in the next research. More case studies and applications of hydrological models can be performed in the future. They are expected to yield insights into the predictive performance of hydrological models."

**Abstract section:**
Line 11: remove "however". "received little attention" is not true if it refers to constant parameters. If it refers to dynamic parameters this is true, but then repeat "dynamic" to make it clear.
**Reply:** Thanks for the referee's reminding. The revision has been completed, as suggested.

Line 15: "probability distributions of violin plots" is an awkward formulation. "probability distributions visualized in violin plots" would be more accurate.
**Reply:** Thanks for the advice. The revision has been done, as suggested.

Line 21 "response [. . .] is generally poor": not clear what this means. Poor in what sense?
**Reply:** The explanation has been added in the Abstract section, as follows.

"There is generally poor response of seasonal dynamic parameter set to dynamic catchment characteristics. Namely, the dynamic changes of parameters did not follow the dynamics of catchment characteristics."

**Text section:**
Pg2 Line 4: "for dynamic" and static "parameters in hydrological" would be more accurate.

**Reply:** Thanks, it has been corrected.

Pg2 Line 14: did you mean: "the more local optima there are"?
**Reply:** Yes, revision has been made, as suggested.

Pg2 Line 32: How does this sentence relate to the previous part of the paragraph? The meaning of the sentence is not clear. What does "measurement of water resources problems" mean?
**Reply:** Thanks for the comment. The ambiguous sentence has been rewritten as follows. "However, the simple and practical method for hydrological modeling with seasonal dynamic parameters still needs to be further explored."

Pg7 Line 15-22: The authors suggest that the width of the parameter distribution is only a function of the number of local minima. This is not true, wide (or "flat") distributions can also be unimodal, in that case, the parameters are just relatively insensitive to the objective function. This needs to be clarified throughout the manuscript (e.g. also on Pg 12, Line 12-13).
**Reply:** Thanks, we have clarified that the probability distributions in violin plots are used to visualize the evolutionary processes by characterizing the structures of fitness landscapes in this study. With an adequate parameter space, namely the same number of iterations, wide (or "flat") distributions might not be relatively unimodal compared with other parameters in a hydrological model.

Pg9 Line 17: The authors repeatedly speak of "the divergence measure". This sounds like there is a quantitative measure to describe the "divergence", which I assume is a measure of the width of a distribution. The term "divergence" is, however, confusing since it has been commonly used for the difference between two distributions, which here is not the case. I suggest the authors replace "divergence" with "width of the parameter distribution" or "standard deviation". Also, I would not call it "measure" unless it is a quantitative indicator.
**Reply:** Thanks for the Referee's comment and advice. The "the divergence measure" has been replaced with "the width of the parameter set distribution", as suggested.

Pg12 Line 5: I don't understand what the authors want to say with this sentence. Figure 7 is referred as Figure 8 in the manuscript.
**Reply:** The ambiguous statement has been changed to "The parameter $K_s$ presents the thinner distribution of violin plots in all sub-periods". Figure 7 has been revised to Figure 8.

Pg16 Line 17-18: It is not clear what the methods are that were used for this analysis. Please describe in more detail and move it to the methods section if necessary.
**Reply:** Thanks. In the methods section, the experiment (operation II) has been redesigned. In operation II, the linear and nonlinear correlation between parameters is first investigated using MIC. Then, a simple but useful tool, i.e., a scatter plot (Paruolo et al., 2013), is used for identifying the sensitive parameter of hydrological models. Only the sensitive parameter is considered as of potential seasonal dynamic parameter, but other parameters are time-invariant.

Table A1: the lower part ("verification-calibration", I think this is supposed to be the difference between validation and calibration) does not add up for columns "NSE" and "LNSE".

**Reply:** Sorry, this was a mistake in Table A1 that has been removed. The model performance has been presented in Figure 3b (see the reply to the previous question).

**References:**

[revised manuscript text omitted]